# Effects of salbutamol on the kinetics of sevoflurane and the occurrence of early postoperative pulmonary complications in patients with mild-to-moderate chronic obstructive pulmonary disease: A randomized controlled study

Huayong Jiang[1‡], Xiujuan Wu[2‡], Shumei Lian[1], Changfeng Zhang[1], Shuyun Liu[1], Zongming Jiang[1]*

1 Department of Anesthesia, Shaoxing People's Hospital (Shaoxing Hospital, Zhejiang University of School of Medicine), Shaoxing, Zhejiang Province, PR China, 2 Department of Nephrology, Shaoxing People's Hospital (Shaoxing Hospital, Zhejiang University of School of Medicine), Shaoxing, Zhejiang Province, PR China

‡ HJ and XW contributed equally to this work as co-first authors.

* jiangzhejiang120@163.com

## Abstract

Bronchodilators dilate the bronchi and increase lung volumes, thereby improving respiratory physiology in patients with chronic obstructive pulmonary disease (COPD). However, their effects on sevoflurane kinetics remain unknown. We aimed to determine whether inhaled salbutamol affected the wash-in and wash-out kinetics of sevoflurane and the occurrence of early postoperative pulmonary complications (PPCs) in patients with COPD undergoing elective surgery. This randomized, placebo-controlled study included 63 consecutive patients with COPD allocated to the salbutamol (n = 30) and control groups (n = 33). The salbutamol group received salbutamol aerosol (2 puffs of ~200 μg) 30 min before anesthesia induction and 30 min before surgery completion. The control group received a placebo. Sevoflurane kinetics were determined by collecting end-tidal samples from the first breaths at 1, 2, 3, 4, 5, 7, 10, and 15 min before the surgery (wash-in) and after closing the vaporizer (wash-out). PPCs were recorded for 7 days. The salbutamol group had higher end-tidal to inhaled sevoflurane ratios (p<0.05, p<0.01) than the control group, from 3 to 10 min during the wash-in period, but no significant differences were observed during the wash-out period. The arterial partial pressure of oxygen to the fraction of inhaled oxygen was significantly higher in the salbutamol group at 30 (320.3±17.6 vs. 291.5±29.6 mmHg; p = 0.033) and 60 min (327.8±32.3 vs. 309.2±30.5 mmHg; p = 0.003). The dead space to tidal volume ratios at 30 (20.5±6.4% vs. 26.3±6.0%, p = 0.042) and 60 min (19.6±5.1% vs. 24.8±5.5%, p = 0.007) and the incidence of bronchospasm (odds ratio [OR] 0.45, 95% confidence interval [CI] 0.23–0.67, p = 0.023) and respiratory infiltration (OR 0.52, 95% CI, 0.40–0.65, p = 0.017) were lower in the salbutamol group. In patients with COPD, salbutamol accelerates the

**Data Availability Statement:** The full data has been uploaded and can be non-commerically used without restriction.The study protocol has the URL: dx.doi.org/10.17504/protocols.io.btidnka6.

**Funding:** This research was supported by Zhejiang Provincial Natural Science Foundation of China under Grant No: LY19H090006 and LQ18H050006.

**Competing interests:** The authors declare no potential interests exist.

wash-in rate of sevoflurane and decreases the occurrence of postoperative bronchospasm and pulmonary infiltration within the first 7 days.

## Introduction

Chronic obstructive pulmonary disease (COPD), which comprises emphysema, chronic bronchitis, and small airway disease, is a chronic inflammatory condition in which the bronchioles are narrowed. COPD is characterized by a partially irreversible airflow limitation due to both chronic airway inflammation and decreased elastic recoil, which eventually leads to uneven air distribution and lung hyperinflation [1]. It poses a special anesthetic challenge in patients with specific pathophysiologic pulmonary changes. In clinical practice, volatile anesthetics are commonly used for inducing and maintaining anesthesia owing to their bronchodilating properties [2,3]. However, the depth of anesthesia may not reach the desired level at a given time due to gas trapping or dynamic hyperinflation upon induction or during surgery, which may delay recovery after surgery [4]. Moreover, COPD places a burden on medical systems due to the high incidence of postoperative pulmonary complications (PPCs) in these patients [5].

Bronchodilators are commonly used in patients with COPD to reduce airway obstruction and ameliorate the symptoms of breathlessness. Previous studies have shown that the administration of salbutamol aerosol via metered-dose inhalers (200–400 μg; 2–4 puffs) significantly alleviated COPD-associated respiratory symptoms and improved lung volume parameters, such as forced expiratory volume in one second, forced vital capacity, and inspiratory capacity, in patients with stable or severe COPD [6–9], and greatly increased the vital capacity while reducing the residual volume in patients with pulmonary emphysema [10,11]. Moreover, in awake patients with COPD, inhaled salbutamol could decrease dynamic hyperinflation by attenuating the expiratory flow limitation, which indicated an improvement in the ventilation status [12].

The improved lung volumes confer benefits to the ventilation status, optimize volume movement, improve uniform gas distribution, and decrease ventilation/perfusion (V/Q) mismatch. Lung ventilation is an important determinant for the transport of volatile agents to the alveolar-capillary interface and further affects the uptake of these agents into blood and the brain [13,14]. We speculated that the administration of salbutamol will accelerate the bulk transportation of sevoflurane, an inhalation anesthetic, to the alveoli through improved lung volume and uniform gas distribution to the lung, while simultaneously enabling the smooth elimination of sevoflurane from the lung and shortening the recovery time [2]. We also hypothesized that salbutamol would be helpful in controlling sputum expectoration and bronchial constriction via its bronchodilating effect on the airway, which eventually may decrease the occurrence of PPCs [13–15].

Therefore, in this randomized, placebo-controlled study, we aimed to assess the effects of salbutamol aerosol on the wash-in and wash-out of sevoflurane during anesthesia via a non-rebreathing circuit and to determine whether this treatment further affected the recovery profile of patients. We also observed whether salbutamol aerosol could decrease the occurrence of PPCs within the first 7 days. The results of this study may provide evidence for treating patients with COPD in terms of the kinetics of volatile agents and pulmonary complications in clinical practice.

## Materials and methods

### Study design and ethics

The study was approved by the Shaoxing People's Hospital Institutional Review Board (Ethics approval No: 2016; Ethic Review Institution Serial No: 096), and written informed consent was

obtained from all patients or caregivers prior to conducting the study. This study was registered at http://www.Chictr.org.cn (Clinical trial number: ChiCTR-1900022899).

The inclusion criteria were as follows: 1) major abdominal surgeries, with a surgical duration exceeding 2 h; 2) age between 65 and 75 years; 3) long history of smoking; 4) confirmed chronic bronchitis, emphysema, or small airway disease; 5) mild-to-moderate COPD based on preoperative lung function; 6) no upper respiratory tract infection within 2 weeks prior to surgery; and 7) American Society of Anesthesiologists physical status I-II. The exclusion criteria were as follows: 1) exacerbation of COPD; 2) allergies to salbutamol preparations, alcohol, or freon; 3) concomitant pulmonary arterial hypertension (defined as a mean pulmonary arterial pressure > 25 mmHg at rest or >30 mmHg under exertion); 4) cardiac dysfunction or heart failure; 5) refusal to participate; and 6) renal insufficiency (defined as a preoperative blood urea nitrogen level > 10 mmol/L and serum creatinine > 1.5 mg/dL). The definitions and grades for COPD were based on the guidelines of the Global Initiative for Chronic Obstructive Disease [16]. Mild-to-moderate COPD was defined as follows: 1) forced expiratory volume in 1 s > 50% of the predicted value and 2) the ratio of forced expiratory volume in 1 s to forced vital capacity below 70%. Between May 2019 and October 2020, eligible patients were consecutively recruited and randomly allocated to the salbutamol group or the control group in accordance with a computer-generated random number sequence.

## Anesthesia management

No premedication was administered to any patient. Thirty minutes before entering the surgery room, patients in the salbutamol group were administered with salbutamol aerosol (200 μg) via a metered-dose inhaler, whereas those in the control group were given a placebo in the same way. The radial artery was cannulated, and a 16-G intravenous line was inserted via the right internal jugular vein. General anesthesia was induced with propofol emulsion (2.0 mg/kg), sufentanil (0.3–0.4 μg/kg), and cis-atracurium (0.2–0.3 mg/kg). The pressure-controlled ventilation mode was initiated after placing the tracheal tube. The peak airway pressure was set to 35 cmH$_2$O, tidal volume was set to 5–7 mL/kg, and the inspiration-to-expiration ratio was adjusted from 1:2.5 to 1:3.5, with a respiratory rate of 10–12 breaths per minute. The level of auto-positive end-expiratory pressure was monitored using the ventilator and was limited to 10 cmH$_2$O by changing the ventilation parameters. Mild hypercapnia was permitted to tolerate the elevated carbon dioxide levels during the surgery; however, when the pH decreased below 7.2, minute ventilation was increased. During the surgery, the bispectral index (Aspect Medical Systems, Norwood, MA, USA) was used to monitor the depth of anesthesia, which was maintained between 45 and 60 by adjusting the infusion rate of remifentanil and propofol. Before the vaporizer was closed, 2% sevoflurane was administered. Cis-atracurium was continuously infused to optimize intraoperative muscle relaxation. A restrictive intravenous-fluid regimen was used to maintain intraoperative urine output between 0.5 and 1.0 mL/kg/h [17]. Vasoconstrictors (preferentially ephedrine) were administered if the mean arterial pressure decreased below 65 mmHg. Atropine was injected to counteract bradycardia (defined as a heart rate < 45 beats per minute). The timing and doses of these vasoactive agents were at the discretion of the attending anesthesiologists.

The respiration circuit was primed with 2% sevoflurane and then connected to the patient's tracheal tube. Simultaneously, the vaporizer was set at 2%, and the fresh gas flow was set at 2 L/min. End-tidal samples were collected from the first breaths at 1, 2, 3, 4, 5, 7, 10, and 15 min before the surgery. The concentrations of inspiratory (F$_I$) and expiratory (F$_A$) sevoflurane were recorded at the proximal end of the endotracheal tube by using an infrared gas analyzer (RGM5250, Ohmeda, BOC Health Care, Inc., Louisville, CO, USA) [13,14]. The analyzer was

calibrated prior to use in accordance with the manufacturer's instructions. Salbutamol aerosol (200 μg) or a placebo was administered via the respiration circuit 30 min before the completion of surgery. The vaporizer was closed 30 min before the last skin suture was placed, with a fresh gas flow of 4 L/min. Following this, end-tidal samples were collected from the first breaths at 1, 2, 3, 4, 5, 7, 10, and 15 min after the discontinuation of administration. Arterial blood samples were extracted from the patients before induction (0 min) and then at 30 and 60 min after the initiation of the surgery. At the end of the surgery, the patients were transferred to the post-anesthesia care unit, and neostigmine (2 mg) and atropine (1 mg) were administered to reverse residual neuromuscular blockade.

## Randomization and interventions

Patients were randomized in a 1:1 ratio to either the salbutamol group or the control group. Randomization was computer-generated, and each patient was given a unique randomization number (patient code). The block length was four. The investigators who were responsible for assessing the primary endpoints, as well as the anesthesiologists, postoperative care unit nursing staff, and variable assessors, were blinded to study group assignment. However, the staff members who collected data during surgery were aware of the group assignments. The placebo used in this study (normal saline) had an appearance identical to the salbutamol preparation, which minimized any bias. Salbutamol aerosol inhalers and identical placebos (containing normal saline) were consecutively numbered from 1 to 80. Eligible patients were also numbered according to the sequence in which they were enrolled in the study.

## Study outcomes

The primary endpoint of this trial was the detection of the effects of salbutamol aerosol pretreatment on the wash-in and wash-out profiles of sevoflurane within 15 min after the initiation or termination of sevoflurane administration. The $F_A/F_I$ ratio was used to represent the processing of sevoflurane. The secondary outcome was the occurrence of PPCs, which were diagnosed based on criteria published in prior studies [18]. We also assessed the recovery profile of the patients by observing time to extubation and time taken to open eyes, squeeze the observer's hand, and provide the exact date of birth on inquiry.

## Statistical analysis

After a careful search of the literature on the internet, we found no similar study pertaining to the kinetics of sevoflurane in patients with COPD undergoing elective surgery. Therefore, a pilot pretest study was conducted (6 cases in each group) to detect the number of patients required in the study. Salbutamol aerosol inhalation 30 min before anesthesia induction would produce a difference of 0.15±0.03 in the end-tidal concentration of sevoflurane. Using a power of 95% and a two-tailed significance level of 0.05, we calculated that a sample size of 25 patients was required. Accounting for a 10% dropout rate owing to loss of follow-up or incomplete data, we intended to enroll 30 patients in each group.

Statistical analysis was performed using PASW Statistics for Windows, Version 18.0 (SPSS Inc., Chicago, IL, USA). In the present study, the main primary outcome was the $F_A/F_I$ ratio, and the differences at each time point were determined using repeated measurement ANOVA followed by the Holm-Sidak post hoc test. The American Society of Anesthesiologists physical status was compared using the Wilcoxon method, and the chi-squared test was used to detect differences in sex, baseline disease history (hypertension and transient ischemic attack), and PPC parameters between the two groups. Preoperative baseline data, such as age, body mass index, ejection fraction, and preoperative lung function; intraoperative parameters;

postoperative parameters, such as arterial partial pressure of oxygen to fraction of inhaled oxygen ($PaO_2/F_iO_2$), dead space volume/tidal volume ($V_D/V_T$), and recovery parameters were compared using Student's t-test or the Mann-Whitney U test. Continuous parameters are presented as mean ± standard deviation, and categorical data are expressed as n (%). A two-tailed p-value less than 0.05 was considered statistically significant.

## Results

A total of 78 patients undergoing abdominal surgery were enrolled during the study period. Two patients with renal insufficiency, three with pulmonary arterial hypertension, and four with chronic heart failure were excluded from the study. Ultimately, 69 patients were included in the study and were randomly assigned to the two groups. Sixty-three patients were included in the final analysis (Fig 1). Their baseline characteristics and operative data are shown in Table 1. The baseline and surgical characteristics had no influence on the outcome measurement in either group.

### Primary outcome

Fig 2 shows the wash-in and wash-out curve profiles for both groups. The $F_A/F_I$ ratios at 3 min (0.74±0.23 vs. 0.65±0.11, $p = 0.041$), 4 min (0.79±0.23 vs. 0.68±0.14, $p = 0.038$), 5 min (0.98 ±0.21 vs. 0.86±0.18, $p = 0.005$), 7 min (1.08±0.23 vs. 0.97±0.20, $p = 0.007$), and 10 min (1.11 ±0.26 vs. 1.31±0.26, $p = 0.021$) were higher in the salbutamol group than in the control group during the wash-in period. However, no statistically significant intergroup differences were observed in the wash-out curve for sevoflurane from 1 to 15 min after its discontinuation.

### Other outcomes

The results regarding the $PaO_2/F_iO_2$ ratio and changes in dead space are shown in Table 2. The $PaO_2/F_iO_2$ ratios at 30 min (320.3±17.6 vs. 291.5±29.6 mmHg, $p = 0.033$) and 60 min (309.2±30.5 vs. 327.8±32.3 mmHg, $p = 0.003$) were significantly higher in the salbutamol group than in the control group. The $V_D/V_T$ ratios at 30 min (20.5±6.4 vs. 26.3±6.0%, $p =$

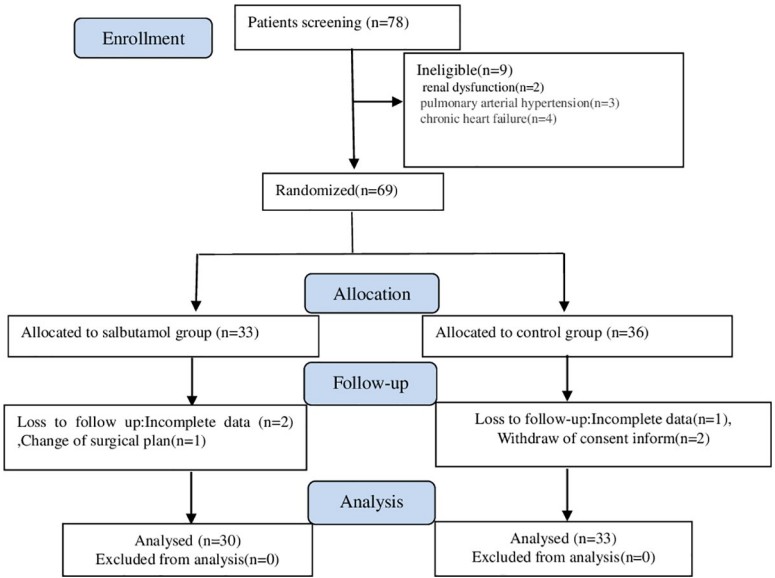

**Fig 1. Consolidated Standards of Reporting Trials (CONSORT) flow chart for this study.**

**Table 1. Baseline characteristics and operative data of the patients.**

| Variable | Control group (n = 33) | Salbutamol group (n = 30) | P-value |
|---|---|---|---|
| **Preoperative[a]** | | | |
| Age (years) | 71.2±6.7 | 70.7±5.9 | 0.641 |
| BMI (kg/m$^2$) | 25.2±2.8 | 24.6±2.5 | 0.712 |
| ASA (I/II)[b] | 15/18 | 12/18 | 0.235 |
| Sex (M/F)[c] | 20/13 | 18/12 | 0.461 |
| Preoperative lung function | | | |
| $FEV_1$, %pred | 62.2±5.7 | 61.3±4.3 | 0.552 |
| FVC, %pred | 82.9±7.9 | 83.4±8.2 | 0.450 |
| $FEV_1$/FVC, % | 54.7±3.6 | 56.3±5.7 | 0.672 |
| DLco, %pred | 90.4±7.5 | 92.7±6.6 | 0.441 |
| Medications | | | |
| Inhaled corticosteroid, n (%) | 15 (45.5%) | 11 (37.8%) | 0.150 |
| Oral corticosteroid, n (%) | 4 (13.7%) | 5 (17.4%) | 0.344 |
| Inhaled bronchodilator, n (%) | 7 (21.2%) | 6 (20%) | 0.561 |
| Combined preparations, n (%) | 12 (35.3%) | 11 (37.1%) | 0.143 |
| EF, (%) | 65.7±4.4 | 66.8±3.7 | 0.987 |
| Hypertension, n (%)[c] | 17 (51.5%) | 15 (50%) | 0.678 |
| TIA, n (%)[c] | 5 (15.5%) | 7 (23.3%) | 0.411 |
| **Intraoperative[a]** | | | |
| Length of anesthesia (min) | 135±62 | 142±71 | 0.132 |
| Duration of surgery (min) | 118±58 | 125±73 | 0.667 |
| Blood loss per hour (mL) | 78±22 | 72±24 | 0.703 |
| Fluids per hour (mL) | 572±88 | 612±87 | 0.451 |
| Urine output per hour (mL) | 141±34 | 154±38 | 0.311 |

Data are presented as the mean ± standard deviation unless specified otherwise.

ASA, American Society of Anesthesiologists physical status; BMI, body mass index; DLco, diffusion capacity of the lung for carbon dioxide; EF, ejection fraction; $FEV_1$, forced expiratory volume in 1 s; FVC, forced vital capacity; $S_pO_2$, oxygen saturation; TIA, transient ischemic attack; %pred, percentage predicted.

[a]p-values were calculated using Student's t-test or the Mann-Whitney U test.

[b]p-value was calculated using the Wilcoxon method.

[c]p-values were calculated using the chi-squared method.

0.042) and 60 min (19.6±5.1 vs. 24.8±5.5%, $p$ = 0.007) were notably lower in the salbutamol group than in the control group.

Table 3 shows the recovery parameters and postoperative pulmonary outcomes. No differences were found in the recovery parameters between the two groups. The development of lung infiltration within 7 days following surgery was more frequent in the control group (9 patients, 27.3%) than in the salbutamol group (2 patients, 6.7%). The frequency of bronchospasm was lower in the salbutamol group (2 patients, 6.7%) than in the control group (9 patients, 27.3%). However, no significant differences were found in the occurrence of atelectasis, pulmonary infection, pneumothorax, and in the length of hospital stay.

## Discussion

The present study on patients with mild-to-moderate COPD demonstrated that (1) the rate of wash-in for sevoflurane increased when salbutamol aerosol was administered preoperatively; (2) salbutamol used perioperatively improved oxygenation by increasing the $PaO_2/F_iO_2$ ratio and reducing the proportion of dead space ventilation; (3) salbutamol administration did not

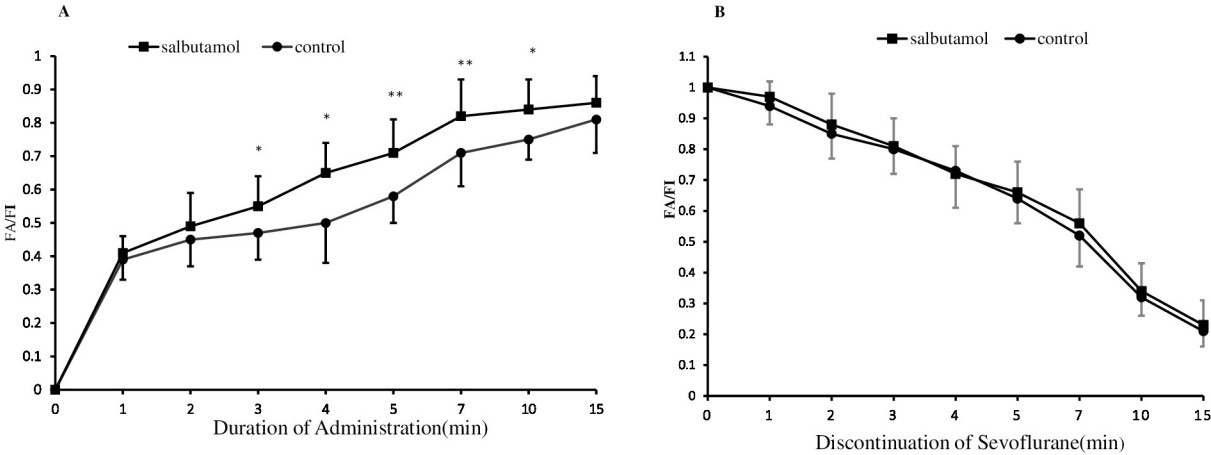

**Fig 2. Wash-in and elimination curves for sevoflurane.** (A) The wash-in profiles for sevoflurane in the salbutamol and control groups after opening the vaporizer ($F_A/F_I$ represents the fraction of end-tidal anesthetic concentration [$F_A$] to the inspired sevoflurane concentration [$F_I$]). (B) The elimination curves for sevoflurane in the salbutamol and control groups after discontinuation 15 min before the end of surgery ($F_A/F_I$ represents the ratio of the end-tidal sevoflurane concentration [$F_A$] to the $F_A$ immediately before the beginning of elimination [$F_{A0}$]). *$p<0.05$, **$p<0.01$ vs. the control group. p-values were calculated using repeated measurement ANOVA followed by the Holm-Sidak post hoc test.

produce clinically relevant differences in the patients' recovery profiles; and (4) salbutamol conferred beneficial effects on pulmonary complications, resulting in the decreased occurrence of bronchospasm and respiratory infiltration in the initial 7 days after surgery.

COPD is characterized by inflammatory small airway disease and parenchymal destruction, which eventually leads to dynamic hyperinflation, nonuniform ventilation, gas trapping, and V/Q mismatch. The accompanying pulmonary function changes impede bulk volume movement and impair gas distribution and exchange under mechanical ventilation. To our knowledge, this is the first study to investigate the effects of bronchodilators on the kinetics of sevoflurane in patients with COPD. After administering salbutamol 30 min before anesthetic induction, the $F_A/F_I$ ratio, which mirrored the wash-in of sevoflurane, from 3 min to 10 min ($p<0.05$ and $p<0.01$, respectively), was higher in the salbutamol group than in the control group. This phenomenon can be attributed to the bronchodilating properties of salbutamol with characteristics of rapid onset [19,20], ameliorated endotracheal intubation-induced bronchoconstriction, improved gas delivery, and enabled homogenous lung ventilation, which might have shortened the time required for sevoflurane to reach the alveoli during the wash-in period. Unfortunately, the respiratory mechanics and real-time gas movement and distribution were not monitored in the present study. Contrary to the wash-in period, the wash-out period showed no significant intergroup differences from 1 to 15 min after the termination of sevoflurane administration. This discrepancy is attributed to the following two factors. First, as most volatile anesthetics (except for desflurane) and most intravenous agents have excellent bronchodilating properties, the airways are maximally relaxed during anesthesia. Second, salbutamol administration 30 min before the conclusion of surgery could not provide any further airway dilation effect and thus, could not produce a clinically relevant difference.

The present results conformed to those of the previous studies [13,14]. In 2014, a study conducted by da Costa et al. [21] demonstrated that the effects of bronchodilation were greater in patients in the initial stage of COPD than in the healthy volunteers and that the improvement in the oscillatory mechanics decreased with increasing severity of COPD, implying that the stages of COPD affected the efficacy of salbutamol. These findings also highlight the fact that salbutamol could not further dilate the bronchus in the context of volatile agent inhalation

Table 2. Comparison of $PaO_2/F_iO_2$ and $V_D/V_T$ between the control and salbutamol groups.

| | 0 min | 30 min | 60 min |
|---|---|---|---|
| **$PaO_2/F_iO_2$ (mmHg)[a]** | | | |
| Control group (n = 33) | 269.0±42.2 | 291.5±38.2 | 309.2±30.5 |
| Salbutamol group (n = 30) | 271.5±36.9 | 320.3±40.1* | 327.8±32.3** |
| **P-value** | 0.132 | 0.033 | 0.003 |
| **$V_D/V_T$ (%)[a]** | | | |
| Control group (n = 33) | 25.4±5.2 | 26.3±6.0 | 24.8±5.5 |
| Salbutamol group (n = 30) | 26.2±6.4 | 20.5±6.4* | 19.6±5.1** |
| **P-value** | 0.245 | 0.042 | 0.007 |

Data are presented as the mean ± standard deviation.

$F_iO_2$, fraction of inspired oxygen; $PaO_2$, arterial partial pressure of oxygen; $V_D$, dead space volume; $V_T$, tidal volume.

*p<0.05 and

**p < 0.01 vs. the control group.

[a] p-values were calculated using Student's t-test or the Mann-Whitney U test.

when administered 30 min before the end of surgery. Therefore, no clinical significance was observed from 1 to 15 min during the wash-out period in our study.

Physiological gas exchange occurs only with homogenous ventilation and perfusion. An appropriate V/Q match is a prerequisite for adequate oxygenation. A high V/Q imbalance, caused by luminal narrowing of the pulmonary arterioles and dynamic hyperinflation, is considered the cause of hypoxia and desaturation in COPD [21]. As shown in Table 2, the $PaO_2/F_iO_2$ ratios were significantly higher at 30 and 60 min in the salbutamol group than in the control group. Our results are consistent with previous results [11,22,23]. A double-blind randomized controlled study revealed that co-administration of nebulized heparin and salbutamol improved oxygenation and the elimination of carbon dioxide when compared to baseline values in patients with acute exacerbations of COPD requiring mechanical ventilation [11]. In the current study, the administration of nebulized salbutamol in patients with mild-to-moderate COPD also improved oxygenation. This might be attributed to the salbutamol-induced dilation of the bronchi, which decreased airway resistance and consequently ameliorated the V/Q mismatch [22].

$V_D$ is the portion of each tidal volume that does not partake in gas exchange. $V_D$ is composed of an anatomical dead space and an alveolar dead space. The $V_D/V_T$ ratio may be as high as 0.25–0.30 in normal healthy individuals. Moreover, the destruction of the alveolar-capillary bed often leads to a regional V/Q mismatch in COPD, which sharply increases the $V_D/V_T$ ratio. Similar to the findings of previous studies [24,25], the $V_D/V_T$ ratio in our study decreased from an initial level of 26.2±6.4% to 20.5±6.4% at 30 min, which further decreased to 19.6±5.1% at 60 min after the administration of the salbutamol aerosol. However, it should be noted that the mechanism underlying these results cannot be easily ascribed to the sole action of salbutamol [26]. Salbutamol alleviates bronchoconstriction and attenuates ventilation heterogeneity, thereby helping maintain the V/Q balance; nevertheless, further studies are needed to understand its role because of the multifactorial influences on dead space fraction [27].

In the present study, the recovery profiles, including the patient's time to opening their eyes, squeezing a hand, extubation, and stating date of birth, were not statistically significant. As shown in Table 3, salbutamol had no notable effect on the trajectory of a patient's recovery from anesthesia. In fact, there are multiple determinants of a patient's recovery from

**Table 3. Recovery parameters and postoperative pulmonary outcomes.**

| | Control group (n = 33) | Salbutamol group (n = 30) | P-value | Odds ratio |
|---|---|---|---|---|
| Recovery parameters[a] | | | | |
| Opening eyes (min) | 13.5±2.3 | 14.2±1.7 | 0.19 | - |
| Squeezing hand (min) | 15.2±3.6 | 16.5±2.2 | 0.57 | - |
| Extubation time (min) | 28.3±10.5 | 30.2±12.1 | 0.34 | - |
| Date of birth (min) | 30.2±8.7 | 32.1±9.2 | 0.15 | - |
| Complications[b] | | | | |
| ICU admission, n (%) | 5 (15.2%) | 2 (6.7%) | 0.102 | 7.22 (0.98–45.32) |
| Bronchospasm, n (%) | 9 (27.3%) | 2 (6.7%)* | 0.023 | 0.45 (0.23–0.67) |
| Atelectasis, n (%) | 2 (6%) | 1 (3.3%) | 0.62 | 1.09 (0.75–1.60) |
| New-onset lung infiltration, n (%) | 9 (27.3%) | 2 (6.7%)* | 0.017 | 0.52 (0.40–0.65) |
| Pulmonary infection, n (%) | 3 (9%) | 2 (6.7%) | 0.52 | 0.98 (0.37–2.66) |
| Pneumothorax | 0 | 0 | - | - |
| Length of hospital stay (days) | 6.7 (5–10) | 7.2 (6–11) | 0.15 | 1.13 (0.45–2.74) |

Data are presented as the mean ± standard deviation unless specified otherwise.

ICU, intensive care unit.

*$p < 0.05$ vs. the control group.

[a]p-values were calculated using Student's t-test.

[b]p-values were using the chi-squared test.

anesthesia [28], and the complicated interplay of these factors ultimately determines the patient's recovery profile. The elimination of volatile agents is often prolonged because of air trapping and V/Q inequalities in patients with COPD [29]. To improve gas delivery and hasten recovery, mitigating dynamic hyperinflation and ameliorating bronchospasm might be advantageous. Salbutamol might play a minimal role in the patient recovery profile during the recovery period owing to its effect on improving gas delivery but could not produce a clinically significant change.

We considered PPCs as a composite outcome measure. Based on the European Joint Task Force Guidelines for Perioperative Clinical Outcome Definitions, PPCs included respiratory infection or failure, pleural effusion, atelectasis, pneumothorax, bronchospasm, aspiration pneumonitis/pneumonia, and acute respiratory distress syndrome [30]. The reported incidence of PPCs ranges from 1% to 67%, and the huge variability is most likely due to differing definitions, study design, patient populations, and specifically, the surgical subspecialty and surgeon's technique [31]. In this study, the incidences of bronchospasm and new-onset pulmonary infiltration were both 6.7% (2 patients) in the salbutamol group and 27.3% (9 patients) and 26.3% (8 patients), respectively, in the control group. In fact, the occurrences of both bronchospasm and new-onset pulmonary infiltration were higher than those previously described in the literature [31]. According to a previous study [32], increasing age is a nonmodifiable risk factor for PPCs. In that study, compared to patients aged below 60 years old, those aged 60–69 years had an odds ratio of 2.1 for a PPC (95% confidence interval [CI]: 1.7–2.6), and those aged 70–79 years had an odds ratio of 3.1 (95% CI: 2.1–4.4). Therefore, as the patient age increased from 65 to 75 years, the risk of PPCs also increased in our clinical trial.

Smoking is a modifiable risk factor for PPCs, and smoking cessation before major surgery reduces postoperative complications [33]. Nearly half of the patients in our study were current smokers, and this contributed to the high occurrence of new-onset pulmonary infiltration (Tables 1 and 3). Bronchospasm was identified based on the clinical manifestation of expiratory wheeze and subsequent treatment with bronchodilators. Further, the occurrence of

postoperative bronchospasm was lower in the salbutamol group than in the control group. Inhalation of salbutamol 30 min before the closure of the surgical incision might have exerted a bronchodilating effect that outlasted the time to extubation, thereby decreasing irritable coughing and increasing expectoration.

This study has a few limitations. First, we used an infrared gas analyzer rather than gas chromatography to measure the concentration of sevoflurane. Second, determining the exact dosage that reached the bronchus under the administration of salbutamol aerosol via a metered-dose inhaler was difficult. Third, this study was limited by its small sample size and short duration of observation. Therefore, additional studies are needed to assess the effects of salbutamol on patients' long-term outcomes, especially the occurrence of PPCs. Finally, the results should be carefully extrapolated to cases of severe COPD, because we only enrolled patients with mild-to-moderate COPD in this study. Despite the results achieved in this study, we recommend that further studies be conducted to verify the results and their applications. Some of the future directions are as follows: (1) the time point for inhaling nebulized salbutamol perioperatively and its optimal dose need further clarification to understand their effects on respiratory mechanics; (2) studies should investigate whether the use of nebulized salbutamol decreases the occurrence of PPCs in patients with severe COPD; (3) research should also focus on whether lung-protective ventilation in conjunction with salbutamol aerosol confers a beneficial effect on the occurrence of PPCs during general anesthesia.

In conclusion, the administration of salbutamol accelerates the wash-in of sevoflurane and decreases the occurrence of bronchospasm and new-onset pulmonary infiltration in patients within the initial 7 days after the surgery.

## Supporting information

**S1 Checklist. CONSORT 2010 checklist of information to include when reporting a randomised trial**\*.
(DOC)

**S1 Table. The profiles of the wash-in of sevoflurane within 15 minutes after initiation.**
(DOCX)

**S2 Table. The profiles of the wash-out of sevoflurane after closing vaporizer.**
(DOCX)

**S1 Dataset.**
(RAR)

**S1 File. 2017 Research protocol salbutamol for submitted.**
(DOCX)

**S2 File.**
(PDF)

**S3 File.**
(DOCX)

**S4 File.**
(PDF)

**S5 File.**
(PDF)

**S6 File.**
(PDF)

## Acknowledgments

We thank all patients who participated in this study. We also thank Prof. Xianhe Zheng, PhD, a senior anesthesiologist, for technical help and Prof. Lin Zhang, a statistician at Shaoxing People's Hospital, for her critical review and explanation of the data. We would like to thank Editage (www.editage.com) for English language editing.

## Author Contributions

**Conceptualization:** Shuyun Liu.

**Data curation:** Changfeng Zhang.

**Formal analysis:** Huayong Jiang.

**Funding acquisition:** Zongming Jiang.

**Investigation:** Huayong Jiang, Xiujuan Wu, Shumei Lian, Changfeng Zhang.

**Methodology:** Xiujuan Wu.

**Project administration:** Shuyun Liu.

**Resources:** Zongming Jiang.

**Supervision:** Zongming Jiang.

**Validation:** Xiujuan Wu.

**Visualization:** Shuyun Liu.

**Writing – original draft:** Huayong Jiang.

**Writing – review & editing:** Zongming Jiang.

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
