## [Decision Letter · Decision Letter 0]

30 Dec 2020

PONE-D-20-33323

Effects of salbutamol on the kinetics of volatile sevoflurane and the occurrence of early postoperative pulmonary complications in patients with mild to moderate chronic obstructive pulmonary disease: a randomized controlled study

PLOS ONE

Dear Dr. Jiang,

Thank you for submitting your manuscript to PLOS ONE. After careful consideration, we feel that it has merit but does not fully meet PLOS ONE’s publication criteria as it currently stands. Therefore, we invite you to submit a revised version of the manuscript that addresses the points raised during the review process.

We look forward to receiving your revised manuscript.

Kind regards,

Walid Kamal Abdelbasset, Ph.D.

Academic Editor

PLOS ONE

Journal Requirements:

2. We note that you state in your manuscript that "The study was approved by our institutional review board (registration number ChiCTR1900022899)" however, this appears to be the clinical trial registration number. Please replace this with your ethics approval number stated in your ethics statement (2016伦审第096号)."

3. Please report the dates on which subject recruitment began and the date range in which the study was performed.

4. Please report your numerical p-values in Table 2.

5.Thank you for stating the following in the Funding Section of your manuscript:

"This study was financially supported by a grant from the Zhejiang Natural Science Foundation program (No: LY19H090006)and Shaoxing Key Discipline of Anesthesia (No：2019SZD04)."

 "The funders had no role in study design,data collection and analysis,decision to publish ,or preparation of the manuscript"

6.We note that you have indicated that data from this study are available upon request. PLOS only allows data to be available upon request if there are legal or ethical restrictions on sharing data publicly. For information on unacceptable data access restrictions, please see http://journals.plos.org/plosone/s/data-availability#loc-unacceptable-data-access-restrictions.

Reviewers' comments:

Reviewer's Responses to Questions

**Comments to the Author**

1. Is the manuscript technically sound, and do the data support the conclusions?

Reviewer #1: Yes

Reviewer #2: Yes

Reviewer #3: Partly

2. Has the statistical analysis been performed appropriately and rigorously? 

Reviewer #1: Yes

Reviewer #2: Yes

Reviewer #3: No

3. Have the authors made all data underlying the findings in their manuscript fully available?

Reviewer #1: Yes

Reviewer #2: No

Reviewer #3: No

4. Is the manuscript presented in an intelligible fashion and written in standard English?

Reviewer #1: Yes

Reviewer #2: Yes

Reviewer #3: No

5. Review Comments to the Author

Reviewer #1: Reviewer comments:

Thank you for giving the opportunity to review this article

Please edit the entire manuscript for English grammar and syntax for good presentation and readability.

Abstract:

1. Change the subtitle – introduction to background and purpose to objective.

2. Define Fa/F1 and ratio PaO2.

3. There is controversies in the duration of outcome measurement .(methods and results section)

4. Conclusion should be more precise as per the reports drawn.

Introduction

1. The introduction part is too short and didn’t mention about important points.

2. How come your study is differed from reference 4 and 5?

3. The research question is not formulated with suitable references.

4. The research problem was not justified clearly

5. Define the clinical significance of this review in related to researchers, clinicians and patients.

Methods

6. The duration of study?

7. The selection criteria should be more specific – indication for surgery.(inclusion and exclusion)

8. Mention the allocation procedure.

9. Mention in detail about the procedure of application in placebo group

10. Include the reliability and validity of the outcome measures used

11. Mention the referral study used for calculating the sample size.

12. Mention the software used for doing statistical tests for replication.

Results

13. Mention the causes for elimination of the subjects from the initial screening.

14. Table 1 – arrange the footnotes in alphabetical order.

15. Table 2 – Include the p values.

16. Table 3 – include the significance of p values through asterisk *

17. Mention the reports with 95% CI (Upper and lower limit) for all variables.

18. Include the effect size.

Discussion

19. Elaborate the discussion part and discuss the relations between the variables.

20. Include MCID value for the primary variable and discuss its effects.

21. Future recommendations of the study is missing

22. Conclusion should be precise based on the objective of the study.

23. Revise the references - follow author guidelines.

24. Fig 2a and 2 b not clear and add the y axis title.

Reviewer #2: *** General remarks: ***

In general, the manuscript is well written, and represents research that should be published. However, some minor amendments should be made to bring the statistical methods and reporting up to par.

*** Specific remarks: ***

The t-tests that are performed are a reasonable analysis method. However, a table should be provided that summarizes the results at each time point. This would allow the reader to perform an adjustment for multiple testing if desired (e.g., using the Bonferroni method). If the authors opt to adjust for multiple testing, then unadjusted p-values should still be given.

The protocol describes a planned repeated measures analysis of variance. This should probably still be done, though could be included as a supplemental analysis. In that case, the simple effects can be described at each time point, and correction made for multiple comparisons using a method such as the Tukey-Kramer method. Unadjusted p-values should still be given in this case also.

The statistical methods section is worded poorly. The following edit is given to indicate the problem areas, rather than complaining about each one individually.

The statistical analysis was performed using SPSS software for Windows (version 18, SPSS Inc, Chicago, IL, USA). Variables were summarized using mean (SD), median (interquartile), or frequency (percentage), as appropriate. Between group differences in baseline characteristics and treatment effects on intraoperative and postoperative measurements were assessed for continuous variables using the unpaired Student's t-test or the Mann-Whitney test, depending on assessment of normality. Association with categorical variables with treatment was assessed using Fisher's exact test. Statistical significance was declared at the 0.05 level.

However, since some additional analysis needs to be described, it is recommended to rewrite the statistical methods section. It is probably best to lay out each set of analyses separately in order of presentation in results for clarity, even at the risk of some repetition.

Analyses that are not described include calculation of odds ratios and confidence intervals. If the planned analysis of variance is performed, this should also be described.

For all tables, please include both frequencies and percentages.

Since no medians and IQRs seem to be presented, perhaps the methods section can be shortened somewhat.

Did the authors evaluate the potential effects of baseline and surgical characteristics on the various outcome measures by plotting them? It is not necessary to present these, but this falls under "due diligence" and should be mentioned in the results section, even if only via a simple one-sentence description.

Figure 2: With this presentation of t-test results, it is necessary to include a table that provides the actual p-values and means and standard deviations. The authors do not need to correct for multiple testing in that case, since the reader can do so, as mentioned above. The figure caption also needs to clearly describe the contents of the figures. Are these means and standard deviations? Usual labelling for axes is "VARIABLE NAME (UNITS)" --- if there are no units then that part is optional. For example, the x-axis of Figure 2a should be something like "Duration of Administration (min)".

As another part of due diligence, the authors should review individual plots of these data by subject, to help identify both outlying observations and outlying subjects.

Line 122: Change "this condition" to "this treatment".

Line 160: It is not clear what is meant by this sentence about hypercapnia. Please rephrase it.

Line 257: It is incorrect to say that the "rate of wash-in" was accelerated. The rate is the change over time. If we evaluate at single time points, we can say that the rate was increased.

Lines 285-288: Note that Table 3 shows that the first several recovery indicators were increased with treatment. Please evaluate and discuss whether this could be a real effect, even if not statistically significant. Given that the variance increased, it might be that a single outlier patient affected all of the results. However, the raw data do not appear to be available for evaluation.

Line 320: In general, instead of "at 6.7%", please write "x of n (6.7%) patients" in general for all of these types of descriptions.

Reviewer #3: Paper titled (Effects of salbutamol on the kinetics of volatile sevoflurane and the occurrence of early postoperative pulmonary complications in patients with mild to moderate chronic obstructive pulmonary disease: a randomized controlled study). by Jiang et al. studied the efect of using salbutamol alongside with volatile sevofluran on the kinetic properties of the later in COPD patients. This is a potentially interesting study with a reasonable clinical outcome.

The main concern on this paper is the rationality. What made the authors think that salbutamol may affect the kinetics of anesthesia? This needs strong justification.

Title: can use COPD in title instead of the complete word. to reduce the length of the title

Introduction contains detailed part on COPD, which can be reduced and to fcoucs more on the drugs and knietics & novelty of the current paper.

Statistical analysis needs clarification. please do not say (when appropriate) but kindly clarify each data and why it was analyzed with certain test. the current writing is somehow confusing.

How normality of distribution was checked?

Student t test was paired or non paired?

Table 1 presentation need to be modified for a perfect appearance. it may be divided to more tables.

In statistics: authors talked about medians., which data was this?

Please write under each illustration, the type od data (mean, %, ...etc) statistical test that was used to assess these data.

6. PLOS authors have the option to publish the peer review history of their article (what does this mean?). If published, this will include your full peer review and any attached files.

Reviewer #1: **Yes: **GOPAL NAMBI

Reviewer #2: No

Reviewer #3: **Yes: **Sawsan A Zaitone

---

## [Author Response · Author response to Decision Letter 0]

15 Feb 2021

Dear editor,

Thank you for your work.

We have revised the manuscript point by point according to the reviewers’ opinion, some paragraphs have been totally rewritten. Few questions we rebut and discuss the related points in the response.

The follows exactly showed that points(we did not change or delete the original versions you sent to us):

Thank you for your consideration. I look forward to hearing from you.

Sincerely,

Dr. Zongming Jiang

Professor and supervisor

Department of Anesthesia

Shaoxing People’s Hospital

No 568,North Zhongxing Road, Yuecheng District of Shaoxing City

Shaoxing, Zhejiang 312000, China

Tel: 86-10-575-88229212

Fax:86-10-575-88229226

E-mail: jiangzhejiang120@163.com

Journal Requirements:

2. We note that you state in your manuscript that "The study was approved by our institutional review board (registration number ChiCTR1900022899)" however, this appears to be the clinical trial registration number. Please replace this with your ethics approval number stated in your ethics statement (2016伦审第096号)."

 Answers: We have rewrite in the ethics statement(Ethic approved No:2016 Ethic Review Institution Serial No:096).

3. Please report the dates on which subject recruitment began and the date range in which the study was performed.

 Answers: From May 2019 to October 2020, the eligible patients were consecutively recruited

4. Please report your numerical p-values in Table 2.

 Answers:The P values were provided.

5.Thank you for stating the following in the Funding Section of your manuscript:

"This study was financially supported by a grant from the Zhejiang Natural Science Foundation program (No: LY19H090006)and Shaoxing Key Discipline of Anesthesia (No：2019SZD04)."

 "The funders had no role in study design,data collection and analysis,decision to publish ,or preparation of the manuscript"

 Answers:We have replaced as mentioned above in the corresponding section of the paper.

6.We note that you have indicated that data from this study are available upon request. PLOS only allows data to be available upon request if there are legal or ethical restrictions on sharing data publicly. For information on unacceptable data access restrictions, please see http://journals.plos.org/plosone/s/data-availability#loc-unacceptable-data-access-restrictions.

 Answers: we will provide raw data. 

 With the permission of hospital ethics committee, we decided to provided the raw data to the journal for reproducibility.

Reviewers' comments:

Reviewer's Responses to Questions

Comments to the Author

1. Is the manuscript technically sound, and do the data support the conclusions?

Reviewer #1: Yes

Reviewer #2: Yes

Reviewer #3: Partly

2. Has the statistical analysis been performed appropriately and rigorously? 

Reviewer #1: Yes

Reviewer #2: Yes

Reviewer #3: No

3. Have the authors made all data underlying the findings in their manuscript fully available?

Reviewer #1: Yes

Reviewer #2: No

Reviewer #3: No

4. Is the manuscript presented in an intelligible fashion and written in standard English?

Reviewer #1: Yes

Reviewer #2: Yes

Reviewer #3: No

5. Review Comments to the Author

Reviewer #1: Reviewer comments:

Thank you for giving the opportunity to review this article

Please edit the entire manuscript for English grammar and syntax for good presentation and readability.

Abstract:

1. Change the subtitle – introduction to background and purpose to objective.

We have changed as the follows:

Background Previous studies have demonstrated that bronchodilators can attenuate bronchoconstriction and improve lung volumes, further improve gas movement and distribution in COPD patients. Whether the improvement of gas movement and distribution can affect the kinetic of sevoflurane remains unknown. 

Objective The study was to determine whether inhaled salbutamol aerosol affects the wash-in and wash-out kinetics of sevoflurane, or the occurrence of early postoperative pulmonary complications in COPD patients undergoing elective surgery.

2. Define Fa/F1 and ratio PaO2.

FA/FI is the concentration of end-tidal to inhaled sevoflurane (FA/FI)ratio.

PaO2/Fio2 is the fraction of arterial pressure of oxygen to the fraction of oxygen. 

We have displayed and defined in the manuscript.

3. There is controversies in the duration of outcome measurement .(methods and results section)

I am sorry for the discrepancy, this is the typesetter error. The correct one was :From May 2019 to October 2020. The outcome measurement was re-defined. 

4. Conclusion should be more precise as per the reports drawn.

Conclusion revised as :Salbutamol nebulization administration accelerates the rate of wash-in of sevoflurane in COPD patients. Meanwhile, decreases the occurrence of bronchospasm and pulmonary infiltration for the initial 7 days after surgery. 

Introduction

1. The introduction part is too short and didn’t mention about important points.

This section had been rewritten thoroughly.

2. How come your study is differed from reference 4 and 5? 

The reference 4 and 5 pertaining to the effect of salbutamol on the airway and symptoms in the exacerbated COPD patients and COPD patients with asthma .In our study, the aim was to study the salbutamol exerts on the volumetrics and airway, which further influence the profile of sevoflurane, eventually affects the quality of recovery and PPC occurrence. 

3. The research question is not formulated with suitable references.

We have carefully read and formulated the references, added 8 new ,replace 5, the total references is 15 in the introduction section.

4. The research problem was not justified clearly.

We have re-written it .

5. Define the clinical significance of this review in related to researchers, clinicians and patients.

Here are extract from the introduction to highlight the significance: We speculated that the administration of salbutamol will accelerate the bulk transportation of sevoflurane to the alveoli through improved lung volume and uniform gas distribution to the lung, and at the same time, provide smooth elimination of sevoflurane from the lung and shorten the length of recovery time. We also hypothesized that salbutamol will be helpful for sputum expectoration and bronchial constriction via its bronchodilating effect on the airway, which eventually decreases the occurrence of PPC .

Methods

6. The duration of study?

From May 2019 to October 2020, the eligible patients were consecutively recruited in the study.

7. The selection criteria should be more specific – indication for surgery. (inclusion and exclusion)

The inclusion criteria were as follows: 1) undergone abdominal major surgeries and the duration of operations exceeded 2 hours; 2) age, 65–75 years; 3) long history of smoking; 4) confirmed chronic bronchitis, emphysema, and small airway disease; 5) mild to moderate COPD based on preoperative lung function testing; 6) no upper respiratory tract infection within 2 weeks prior to surgery; 7) American Society of Anesthesiologists physical status 1-2. 

The exclusion criteria were: 1) exacerbation of COPD; 2) allergies to salbutamol preparations, alcohol, or freon; 3) concomitant pulmonary arterial hypertension (defined as mean pulmonary arterial pressure >25 mmHg at rest or >30 mmHg under exertion); 4) cardiac dysfunction or heart failure; 5) refusal to participate; 6) renal insufficiency (defined as preoperative blood urea nitrogen level ＞10 mmol·L-1, serum creatinine ＞1.5 mg·dl-1).

8. Mention the allocation procedure.

Answer: The computer-generated random sequence numbers were stored in an opaque envelope. Randomization was conducted using sealed, sequentially numbered, and opaque envelopes. Patients who satisfied all inclusion criteria and had no exclusion criteria were randomly assigned in a 1:1 ratio to either salbutamol group and control group. The investigators who were responsible for assessing the primary endpoints, as well as the anesthesiologists, postoperative care unit nursing staff, and variable assessors, were blinded to study group assignment. We chose to use salbutamol and its identical in appearance placebo (normal saline) in this study.

We have described the allocation process in the manuscript.

9. Mention in detail about the procedure of application in placebo group

We chose to use salbutamol and its similar in appearance placebo (normal saline) in this study. In placebo group, its identical placebo was used in the same manner to that in the salbutamol group. 

10. Include the reliability and validity of the outcome measures used

(1) End-tidal sevoflurane , as measured with the anesthesia gas monitor. The device incorporated into the anesthesia machine, is a standard device. Before the initiation of every patients, the device was calibrated according to the instruction. The sampling line was disposable and discard after use to minimize the water gathering in the line, which will influence the accuracy of the results. 

(2) The sidestream method (sampling 0.3 ml gas from the circuit) was used to measure the fraction of sevoflurane in the admixture of gas in the respiratory circuit. 

(3) Drager primus anesthesia machine was selected in the study to decrease and minimize the difference from the machine.

(4) The vaporizer(2%) and oxygen flow(2L/min) was fixed in each measurement.

(5)The determination of kinetic profiles of inhalational agents using analysis of end-tidal gas concentration has already been obtained for other anesthetic gases, such as halothane and isoflurane. In this study, we used those data which are routinely available from anesthetic workstations. Therefore, we ensure the reliability of the each measurement (FA,FI). You can refer the literature : Rietbrock S, Wissing H, Kuhn I, Fuhr U. Pharmacokinetics of inhaled anaesthetics in a clinical setting: evaluation of a method based on routine monitoring data. Br J Anaesth2000:84:437-42.

11. Mention the referral study used for calculating the sample size.

After careful search of the literature on the internet, we found out that no similar study pertaining to the kinetics of sevoflurane in COPD patients undergoing elective surgery and so a pilot pretest study was conducted ( 6 cases in each group) to detect the number of subjects required in the study. In the presence of salbutamol aerosol inhalation 30 minutes before anesthesia induction would produce a 0.15�0.03 difference of end-tidal concentration of sevoflurane. Using a power of 95% and a two-tailed significance level of 0.05, we calculated that a sample size of 25 patients was required. Accounting for a 10% dropout rate due to loss of follow-up or incomplete data, we intended to enroll 30 patients in each group. It should be pointed out that 80 random number was generated and enveloped before the conduction of study.

12. Mention the software used for doing statistical tests for replication.

The statistical analysis was performed using SPSS software for Windows (version 18, SPSS Inc, Chicago, IL, USA).

Results

13. Mention the causes for elimination of the subjects from the initial screening.

2 cases with renal insufficiency, 3 patients with pulmonary arterial hypertension and 4 with chronic heart failure were eliminated from the study. Ultimately, 69 cases were included in the study and were randomly assigned to the two groups. Sixty-three cases were included in the final analysis (Fig 1).

14. Table 1 – arrange the footnotes in alphabetical order.

We have re-arranged like this, Footnotes: ASA, American Society of Anesthesiologists ; BMI, body mass index;DLco, diffusion capacity of lung for carbon dioxide; EF, ejection fraction; FEV1, forced expiratory volume in 1s; FVC, forced vital capacity; SpO2, oxygen saturation; TIA, transient ischemic attack. Values are presented as mean±SD, number or percentage of patients.

15. Table 2 – Include the p values.

It was added in the table.

16. Table 3 – include the significance of p values through asterisk *

It was added in the table.

17. Mention the reports with 95% CI (Upper and lower limit) for all variables.

It was added the odds ratio with 95% CI in table3.

18. Include the effect size. 

(0.15�0.03)difference of end-tidal concentration of sevoflurane in the pretest. And this is the basis for computing the sample size.

19. Elaborate the discussion part and discuss the relations between the variables.

We have re-write the manuscript based on the suggestions.

20. Include MCID value for the primary variable and discuss its effects.

The minimal clinically important difference (MICD) in the study, a 0.05-0.07 FA/FI difference convert to the (0.15�0.03) difference of end-tidal concentration of sevoflurane according to literature (La Colla G, La Colla L, Turi S, Poli D, Albertin A, Pasculli N, Bergonzi PC, Gonfalini M, Ruggieri F. Effect of morbid obesity on kinetic of desflurane: wash-in wash-out curves and recovery times. Minerva Anestesiol. 2007 May;73(5):275-9. PMID: 17529920.) . And the above is based on salbutamol on lung volume in different state , so through the changes in lung volume, it will affect the kinetics of sevoflurane. The references as follows:

(1)Al-saady N，Bennett ED. Decelerating inspiratory flow waveform improves lung mechanics and gas exchange in patients on intermittent positive pressure ventilation[J]. Intensive care med，1985，11: 68-75. (2)Shieh CY，Sye PY. Effects of inspiratory flow waveforms on lung mechanics and gas exchange and respiratory metabolism in COPD patients during mechanical ventilation[J]. Chest，2002, 122: 2096-2104.

(3)Tantucci C， Duguet A， Similowski T，et al. Effect of salbutamol on dynamic hyperinflation in chronic obstructive pulmonary disease patients[J]. Eur Respir J，1998，12: 799-804. 

(4)Newton MF, O'Donnell DE, Forkert L. Response of Lung Volumes to Inhaled Salbutamol in a Large Population of Patients With Severe Hyperinflationm [J]. Chest, 2002, 121(4): 1042-1050. 

 The table described the lung volume change based on the above references.

 Therefore ,in the discussion of the manuscript, we did not further discuss the MCID . 

21. Future recommendations of the study is missing

The prospective study concerning time and optimal dose for inhaling salbutamol perioperatively is further needed to clarify the effects on respiratory mechanics, which confers benefit to the PPC, in COPD patients under various ventilation mode.

22. Conclusion should be precise based on the objective of the study.

The conclusion of the manuscript was changed to this : In conclusion, the administration of salbutamol accelerates the wash-in of sevoflurane and decreases the occurrence of bronchospasm and new-onset pulmonary infiltration in patients within the initial 7 days after surgery.

23. Revise the references - follow author guidelines.

Have been carefully revised.

24. Fig 2a and 2 b not clear and add the y axis title.

The resolution was changed based on journal criteria.

Reviewer #2: *** General remarks: ***

In general, the manuscript is well written, and represents research that should be published. However, some minor amendments should be made to bring the statistical methods and reporting up to par.

*** Specific remarks: ***

The t-tests that are performed are a reasonable analysis method. However, a table should be provided that summarizes the results at each time point. This would allow the reader to perform an adjustment for multiple testing if desired (e.g., using the Bonferroni method). If the authors opt to adjust for multiple testing, then unadjusted p-values should still be given.

The protocol describes a planned repeated measures analysis of variance. This should probably still be done, though could be included as a supplemental analysis. In that case, the simple effects can be described at each time point, and correction made for multiple comparisons using a method such as the Tukey-Kramer method. Unadjusted p-values should still be given in this case also.

The statistical methods section is worded poorly. The following edit is given to indicate the problem areas, rather than complaining about each one individually.

The statistical analysis was performed using SPSS software for Windows (version 18, SPSS Inc, Chicago, IL, USA). Variables were summarized using mean (SD), median (interquartile), or frequency (percentage), as appropriate. Between group differences in baseline characteristics and treatment effects on intraoperative and postoperative measurements were assessed for continuous variables using the unpaired Student's t-test or the Mann-Whitney test, depending on assessment of normality. Association with categorical variables with treatment was assessed using Fisher's exact test. Statistical significance was declared at the 0.05 level.

 Honestly, the section of statistical analysis caused the confusion. The statistical methods section worded thoroughly.

After careful search of the literature on the internet, we found out that no similar study pertaining to the kinetics of sevoflurane in COPD patients undergoing elective surgery and so a pilot pretest study was conducted ( 6 cases in each group) to detect the number of subjects required in the study. In the presence of salbutamol aerosol inhalation 30 minutes before anesthesia induction would produce a 0.15�0.03difference of end-tidal concentration of sevoflurane. Using a power of 95% and a two-tailed significance level of 0.05, we calculated that a sample size of 25 patients was required. Accounting for a 10% dropout rate due to loss of follow-up or incomplete data, we intended to enroll 30 patients in each group.

The statistical analysis was performed using SPSS software for Windows (version 18, SPSS Inc, Chicago, IL, USA).The main cutoff value was the FA/FI ratio, deriving from concentrations of expiratory (FA) and inspiratory (FI) fraction, the difference at each time point were determined by Student’s t-test. Variables were presented as mean� standard deviation (SD) or number (percentage) as appropriate. 

The baseline demographic data, and intraoperative and postoperative measurements, including PaO2/FiO2 ,VD/VT and recovery parameters were tested for homogeneity of variance, were compared by the student t-test or the Mann-Whitney test. As for the components of PPC, Pearson Chi-square test was used to compare difference between two groups. A two -tailed P value less than 0.05 was considered statistically significant.

Moreover, from the suggestion of a senior statistician ,it is not necessary for performing the Bonferroni test among different timepoints between the groups because the main outcome(FA/FI ratio) in each time point is obvious different based the physiology of the kinetic of volatile agents.

However, since some additional analysis needs to be described, it is recommended to rewrite the statistical methods section. It is probably best to lay out each set of analyses separately in order of presentation in results for clarity, even at the risk of some repetition.

We have re-written it .

Analyses that are not described include calculation of odds ratios and confidence intervals. If the planned analysis of variance is performed, this should also be described.

We have re-written it .

For all tables, please include both frequencies and percentages.

We have re-written it .

Since no medians and IQRs seem to be presented, perhaps the methods section can be shortened somewhat.

We have re-written it .

Did the authors evaluate the potential effects of baseline and surgical characteristics on the various outcome measures by plotting them? It is not necessary to present these, but this falls under "due diligence" and should be mentioned in the results section, even if only via a simple one-sentence description.

We have re-written it like “The baseline and surgical characteristics conferred no influence on outcome measurement in both groups.”

Figure 2: With this presentation of t-test results, it is necessary to include a table that provides the actual p-values and means and standard deviations. The authors do not need to correct for multiple testing in that case, since the reader can do so, as mentioned above. The figure caption also needs to clearly describe the contents of the figures. Are these means and standard deviations? Usual labelling for axes is "VARIABLE NAME (UNITS)" --- if there are no units then that part is optional. For example, the x-axis of Figure 2a should be something like "Duration of Administration (min)".

FA/FI ratio had no units because it was the fraction of end-tidal samples of the sevoflurane (concentrations of expiratory (FA) and inspiratory (FI) fraction).A table that provides the actual p-values and means and standard deviations were included in the figures.

As another part of due diligence, the authors should review individual plots of these data by subject, to help identify both outlying observations and outlying subjects.

We performed this, this is not necessary for displaying in the manuscript. 

Line 122: Change "this condition" to "".

We have revised as “this treatment” be more precise and be apt for .

Line 160: It is not clear what is meant by this sentence about hypercapnia. Please rephrase it.

Hypercapnia is a phrase described the level of arterial pressure of carbon dioxide , the normal level is 35-45 mmHg in normal healthy patients, In COPD, the level often higher due to the impaired respiratory function, which called hypercapnia.

Line 257: It is incorrect to say that the "rate of wash-in" was accelerated. The rate is the change over time. If we evaluate at single time points, we can say that the rate was increased.

I agree with the reviewer’s opinion.

Lines 285-288: Note that Table 3 shows that the first several recovery indicators were increased with treatment. Please evaluate and discuss whether this could be a real effect, even if not statistically significant. Given that the variance increased, it might be that a single outlier patient affected all of the results. However, the raw data do not appear to be available for evaluation.

In present study, the recovery profiles, including a patient’s time to opening their eyes, squeezing a hand, extubation, and stating their own name, were not statistically significant. As shown in Table 3, it appears that salbutamol had no notable effect on the trajectory of a patient’s recovery from anesthesia. In fact, the determinants for a patient’s recovery from anesthesia are multifactorial [28], and the complicated interplay of these factors ultimately determines the patient’s recovery profile. The elimination of volatile agents is often prolonged due to air trapping and V/Q inequalities in patients with COPD[29]. To improve gas delivery and hasten recovery, it is advantageous to mitigate dynamic hyperinflation and ameliorate bronchospasm. Salbutamol might play a minimal role in patient recovery profile during the waking period due to its effect on improving gas delivery whereas could not produce a clinical significance.

Line 320: In general, instead of "at 6.7%", please write "x of n (6.7%) patients" in general for all of these types of descriptions.

We have revised.

Reviewer #3: Paper titled (Effects of salbutamol on the kinetics of volatile sevoflurane and the occurrence of early postoperative pulmonary complications in patients with mild to moderate chronic obstructive pulmonary disease: a randomized controlled study). by Jiang et al. studied the effect of using salbutamol alongside with volatile sevoflurane on the kinetic properties of the later in COPD patients. This is a potentially interesting study with a reasonable clinical outcome.

The main concern on this paper is the rationality. What made the authors think that salbutamol may affect the kinetics of anesthesia? This needs strong justification.

It has rationality and justification. And salbutamol had effect on lung volume in COPD patients. The evidence :

(1)Al-saady N，Bennett ED. Decelerating inspiratory flow waveform improves lung mechanics and gas exchange in patients on intermittent positive pressure ventilation[J]. Intensive care med，1985，11: 68-75. (2)Shieh CY，Sye PY. Effects of inspiratory flow waveforms on lung mechanics and gas exchange and respiratory metabolism in COPD patients during mechanical ventilation[J]. Chest，2002, 122: 2096-2104.

(3)Tantucci C， Duguet A， Similowski T，et al. Effect of salbutamol on dynamic hyperinflation in chronic obstructive pulmonary disease patients[J]. Eur Respir J，1998，12: 799-804. 

(4)Newton MF, O'Donnell DE, Forkert L. Response of Lung Volumes to Inhaled Salbutamol in a Large Population of Patients With Severe Hyperinflationm [J]. Chest, 2002, 121(4): 1042-1050. 

 The table described the lung volume change based on the above references.

 parameters Severe inflation (mL) Moderate inflation (mL)

Decreased� TLC

FRC

RV 222±13

442±26

510±28 150±10

260±39

300±14

Increased� IC

FVC

FEV1 220±15

336±21

160±10 110±11

204±13

150±10

Reaction flow

volume 33%

76% 26%

62%

Newton MF, O'Donnell DE, Forkert L. Response of lung volumes to inhaled salbutamol in a large population of patients with severe hyperinflation. Chest. 2002 Apr;121(4):1042-50. doi: 10.1378/chest.121.4.1042.

La Colla G, La Colla L, Turi S, Poli D, Albertin A, Pasculli N, Bergonzi PC, Gonfalini M, Ruggieri F. Effect of morbid obesity on kinetic of desflurane: wash-in wash-out curves and recovery times. Minerva Anestesiol. 2007 May;73(5):275-9. PMID: 17529920.

Weingarten TN, Hawkins NM, Beam WB, Brandt HA, Koepp DJ, Kellogg TA, Sprung J. Factors associated with prolonged anesthesia recovery following laparoscopic bariatric surgery: a retrospective analysis. Obes Surg. 2015 Jun;25(6):1024-30. doi: 10.1007 Lee SH, Kim N, Lee CY, Ban MG, Oh YJ. Effects of dexmedetomidine on oxygenation and lung mechanics in patients with moderate chronic obstructive pulmonary disease undergoing lung cancer surgery: A randomised double-blinded trial. Eur J Anaesthesiol. 2016 Apr;33(4):275-82. doi: 10.1097/EJA.0000000000000405. PMID: 26716866;/s11695-014-1468-7. PMID: 25392076; 

Title: can use COPD in title instead of the complete word. to reduce the length of the title

Introduction contains detailed part on COPD, which can be reduced and to fcoucs more on the drugs and knietics & novelty of the current paper.

It has been totally revised in introduction section.

Statistical analysis needs clarification. please do not say (when appropriate) but kindly clarify each data and why it was analyzed with certain test. the current writing is somehow confusing.

How normality of distribution was checked?

Student t test was paired or non paired?

The section has been changed:

After careful search of the literature on the internet, we found out that no similar study pertaining to the kinetics of sevoflurane in COPD patients undergoing elective surgery and so a pilot pretest study was conducted ( 6 cases in each group) to detect the number of subjects required in the study. In the presence of salbutamol aerosol inhalation 30 minutes before anesthesia induction would produce a 0.15�0.03difference of end-tidal concentration of sevoflurane. Using a power of 95% and a two-tailed significance level of 0.05, we calculated that a sample size of 25 patients was required. Accounting for a 10% dropout rate due to loss of follow-up or incomplete data, we intended to enroll 30 patients in each group.

The statistical analysis was performed using SPSS software for Windows (version 18, SPSS Inc, Chicago, IL, USA).The main cutoff value was the FA/FI ratio, deriving from concentrations of expiratory (FA) and inspiratory (FI) fraction, the difference at each time point were determined by Student’s t-test. Variables were presented as mean� standard deviation (SD) or number (percentage) as appropriate. 

The baseline demographic data, and intraoperative and postoperative measurements, including PaO2/FiO2 ,VD/VT and recovery parameters were tested for homogeneity of variance, were compared by the student t-test or the Mann-Whitney test. As for the components of PPC, Pearson Chi-square test was used to compare difference between two groups. A two -tailed P value less than 0.05 was considered statistically significant.

Table 1 presentation need to be modified for a perfect appearance. it may be divided to more tables.

We omitted some parts of the baseline data, which had no close relation to and influence on the primary outcome. And the table were not divided.

In statistics: authors talked about medians., which data was this?

The section has been changed in the paper.

Please write under each illustration, the type od data (mean, %, ...etc) statistical test that was used to assess these data.

The section has been changed in the paper.

6. PLOS authors have the option to publish the peer review history of their article (what does this mean?). If published, this will include your full peer review and any attached files.

Do you want your identity to be public for this peer review? For information about this choice, including consent withdrawal, please see our Privacy Policy.

Reviewer #1: Yes: GOPAL NAMBI

Reviewer #2: No

Reviewer #3: Yes: Sawsan A Zaitone

---

## [Decision Letter · Decision Letter 1]

5 Mar 2021

PONE-D-20-33323R1

Effects of salbutamol on the kinetics of sevoflurane and the occurrence of early PPC in patients with mild to moderate COPD: a randomized controlled study

PLOS ONE

Dear Dr. Jiang,

Thank you for submitting your manuscript to PLOS ONE. After careful consideration, we feel that it has merit but does not fully meet PLOS ONE’s publication criteria as it currently stands. Therefore, we invite you to submit a revised version of the manuscript that addresses the points raised during the review process.

We look forward to receiving your revised manuscript.

Kind regards,

Walid Kamal Abdelbasset, Ph.D.

Academic Editor

PLOS ONE

Reviewers' comments:

Reviewer's Responses to Questions

**Comments to the Author**

1. If the authors have adequately addressed your comments raised in a previous round of review and you feel that this manuscript is now acceptable for publication, you may indicate that here to bypass the “Comments to the Author” section, enter your conflict of interest statement in the “Confidential to Editor” section, and submit your "Accept" recommendation.

Reviewer #1: All comments have been addressed

Reviewer #2: All comments have been addressed

Reviewer #3: (No Response)

2. Is the manuscript technically sound, and do the data support the conclusions?

Reviewer #1: Partly

Reviewer #2: Yes

Reviewer #3: Partly

3. Has the statistical analysis been performed appropriately and rigorously? 

Reviewer #1: Yes

Reviewer #2: Yes

Reviewer #3: No

4. Have the authors made all data underlying the findings in their manuscript fully available?

Reviewer #1: Yes

Reviewer #2: No

Reviewer #3: No

5. Is the manuscript presented in an intelligible fashion and written in standard English?

Reviewer #1: No

Reviewer #2: Yes

Reviewer #3: (No Response)

6. Review Comments to the Author

Reviewer #1: Reviewer comments:

Thank you for giving the opportunity to review this article.

Abstract:

1. Include the acronym of abbreviation when using first time.

Introduction

2. Please check the grammar, syntax and paragraph format.

3. Add the clinical significance of this article over the participants and researchers.

Methods

4. Include the study design, randomization and allocation procedure.

5. Include the reliability and validity of outcome measures with references.

Results

6. Include the reports with CI 95% with p scores..

7. Mention the effect size of primary variable and its MCID scores.

Discussion

8. The discussion part should discuss the relation between the outcome variables with latest references.

9. Include the future recommendations of this study.

Reviewer #2: The authors have essentially dealt with my substantive criticisms of their previous statistical analysis. Even though we might disagree on some minor issues, the choices made by the authors are reasonable and allow the readers to make their own determination if they disagree a little.

My primary current reservations around the manuscript center around the English language usage. There are many small errors in grammar and usage that should probably be addressed, especially in the newly minted statistical methods section.

Reviewer #3: Authors partly revised the Ms. and some of the suggestions were not addressed.

Authors did not mention how they reached the sample size in the methods

Authors need to transparently mention the stst analyses method and type of data in each illustration (in footnotes mention the type of data presented and stat test)

In methods:the difference at each time point were determined by Student’s t-tes (this is not correct to be done, t test is not the correcct test for the situation, all values should be assessed by ANOVA then a post hoc test)

& nothing is mentioned in the footnotes!!

Authors wrote also: were compared by the student t-test or the Mann-Whitney test. : this reviewer feels different from that wrote in the table footnote , what are the basis for selecting any of the 2 tests???

Overall the stat analysis in METHODs is not matched with that in the tables. Footnotes does not include sufficient info about the data

Authors wrote: Pearson Chi-square test was used to compare difference between two groups.

Q: Is there a test called {Pearson Chi-square test}?????

7. PLOS authors have the option to publish the peer review history of their article (what does this mean?). If published, this will include your full peer review and any attached files.

Reviewer #1: **Yes: **Gopal Nambi

Reviewer #2: No

Reviewer #3: No

---

## [Author Response · Author response to Decision Letter 1]

3 Apr 2021

Dear editor,

Thank you for your work.

We have revised the manuscript point by point according to the reviewers’ opinion, some paragraphs have been totally rewritten. Few questions we rebut and discuss the related points in the response.

The follows exactly showed that points(we did not change or delete the original versions you sent to us):

Thank you for your consideration. I look forward to hearing from you.

Sincerely,

Dr. Zongming Jiang

Professor and supervisor

Department of Anesthesia

Shaoxing People’s Hospital

No 568,North Zhongxing Road, Yuecheng District of Shaoxing City

Shaoxing, Zhejiang 312000, China

Tel: 86-10-575-88229212

Fax:86-10-575-88229226

E-mail: jiangzhejiang120@163.com

PONE-D-20-33323R1

Effects of salbutamol on the kinetics of sevoflurane and the occurrence of early PPC in patients with mild to moderate COPD: a randomized controlled study

PLOS ONE

Dear Dr. Jiang,

Thank you for submitting your manuscript to PLOS ONE. After careful consideration, we feel that it has merit but does not fully meet PLOS ONE’s publication criteria as it currently stands. Therefore, we invite you to submit a revised version of the manuscript that addresses the points raised during the review process.

We look forward to receiving your revised manuscript.

Kind regards,

Walid Kamal Abdelbasset, Ph.D.

Academic Editor

PLOS ONE

Answer:We have uploaded the protocols to protocols.io and DOI is:

dx.doi.org/10.17504/protocols.io.btidnka6

Reviewers' comments:

Reviewer's Responses to Questions

Comments to the Author

1. If the authors have adequately addressed your comments raised in a previous round of review and you feel that this manuscript is now acceptable for publication, you may indicate that here to bypass the “Comments to the Author” section, enter your conflict of interest statement in the “Confidential to Editor” section, and submit your "Accept" recommendation.

Reviewer #1: All comments have been addressed

Reviewer #2: All comments have been addressed

Reviewer #3: (No Response)

2. Is the manuscript technically sound, and do the data support the conclusions?

Reviewer #1: Partly

Reviewer #2: Yes

Reviewer #3: Partly

3. Has the statistical analysis been performed appropriately and rigorously? 

Reviewer #1: Yes

Reviewer #2: Yes

Reviewer #3: No

4. Have the authors made all data underlying the findings in their manuscript fully available?

Reviewer #1: Yes

Reviewer #2: No

Reviewer #3: No

5. Is the manuscript presented in an intelligible fashion and written in standard English?

Reviewer #1: No

Reviewer #2: Yes

Reviewer #3: (No Response)

6. Review Comments to the Author

Reviewer #1: Reviewer comments:

Thank you for giving the opportunity to review this article.

Abstract:

1. Include the acronym of abbreviation when using first time.

Answer: It has been changed like this:chronic obstructive pulmonary disease (COPD) patients.

Introduction

2. Please check the grammar, syntax and paragraph format.

Answer: we have re-formatted and check the gramma.

3. Add the clinical significance of this article over the participants and researchers.

Answer: The significance of our current study is that the results of the study will provide evidence for treating patients with COPD as regards to kinetics of volatiles and pulmonary complications in clinical practice.

Methods

4. Include the study design, randomization and allocation procedure.

Answer: Patients were randomized in a 1:1 ratio to either the salbutamol group or the control group. Randomization was computer-generated and each patient was given a unique randomization number (patient code).The block length was four. The investigators who were responsible for assessing the primary endpoints, as well as the anesthesiologists, postoperative care unit nursing staff, and variable assessors, were blinded to study group assignment. The staff members who collected data during surgery were aware of the group assignments. The salbutamol preparations and its identical in appearance placebo (normal saline) were used to minimize the bias in this study. Salbutamol aerosol inhalers and its identical placebo (containing normal saline) were consecutively numbered from 1 to 80. And the eligible patients were also numbered based on sequence enrolled into the study. 

5. Include the reliability and validity of outcome measures with references.

Answer: The outcome measures : The FA/FI ratio was assessed to represent the processing of sevoflurane. FA and FI are extracted from the anesthesia machine, which has module to monitor and measure. Before each experiment, we will calibrate it according to the instructions. It is reliable. Determinations of other parameters were based on the standard definitions, for example PPC.

Results

6. Include the reports with CI 95% with p scores..

Answer: It has been analyzed and added.

7. Mention the effect size of primary variable and its MCID scores.

Answer: Bronchodilators produce bronchodilation and increase lung volumes, improves volume bulk movement in patients withCOPD. Effect size in our study is the fraction difference of sevoflurane in the admixture of inspired gas, the clinical significance is reflecting the wash-in of volatiles, a faster wash-in produces shorter time to the desired depth of anesthesia. If in awake patients, there will be difference in time of loss of consciousness. The MCID is difficult to determine, and often based on experts’ opinion or clinical experience in the current study. According to the literature:Torri G, Casati A, Comotti L, Bignami E, Santorsola R, Scarioni M. Wash-in and wash-out curves of sevoflurane and isoflurane in morbidly obese patients. Minerva Anestesiol. 2002 Jun;68(6):523-7. PMID: 12105408. The minimal clinically important difference(MCID) is 0.05 to 0.08 of exhaled sevoflurane concentration, which often produce a little faster time of loss of consciousness (15-22seconds).In fact ,it is a complicated interplay between respiration and cardiovascular system. So, we did not clearly point out the effect size of primary variable and its MCID scores in the manuscript. As in the paper :Sharma P, Gombar S, Ahuja V, Jain A, Dalal U. Sevoflurane sparing effect of dexmedetomidine in patients undergoing laparoscopic cholecystectomy: A randomized controlled trial. J Anaesthesiol Clin Pharmacol. 2017 Oct-Dec;33(4):496-502. doi: 10.4103/joacp.JOACP_144_16. PMID: 29416243; PMCID: PMC5791264.

Discussion

8. The discussion part should discuss the relation between the outcome variables with latest references.

Answer:We have related the current results to the previous studies.

9. Include the future recommendations of this study.

Answer: we recommend like this as the followings: Despite the results achieved in the study, we recommended that further studies should be conducted to testify the results and its applications. As follows: (1) the time and optimal dose for inhaling nebulized salbutamol perioperatively is further needed to clarify the effects on respiratory mechanics; (2) whether the use of nebulized salbutamol decreases the PPC occurrence in patients with severe COPD.(3) whether lung protective ventilation in conjunction with salbutamol aerosol confers beneficial effect on PPC during general anesthesia.

Reviewer #2: The authors have essentially dealt with my substantive criticisms of their previous statistical analysis. Even though we might disagree on some minor issues, the choices made by the authors are reasonable and allow the readers to make their own determination if they disagree a little.

My primary current reservations around the manuscript center around the English language usage. There are many small errors in grammar and usage that should probably be addressed, especially in the newly minted statistical methods section.

Answer: Language had been proof through technical assistance.

Reviewer #3: Authors partly revised the Ms. and some of the suggestions were not addressed.

Authors did not mention how they reached the sample size in the methods

Answer: We draw the sample size as follows: After careful search of the literature on the internet, we found out that no similar study pertaining to the kinetics of sevoflurane in COPD patients undergoing elective surgery and so a pilot pretest study was conducted (6 cases in each group) to detect the number of subjects required in the study. In the presence of salbutamol aerosol inhalation 30 minutes before anesthesia induction would produce a difference of 0.15�0.03 in end-tidal concentration of sevoflurane. Using a power of 95% and a two-tailed significance level of 0.05, we calculated that a sample size of 25 patients was required. Accounting for a 10% dropout rate due to loss of follow-up or incomplete data, we intended to enroll 30 patients in each group.

Authors need to transparently mention the stst analyses method and type of data in each illustration (in footnotes mention the type of data presented and stat test)

Answer: we added and stated the detailed analyses and footnotes.

In methods:the difference at each time point were determined by Student’s t-tes (this is not correct to be done, t test is not the correcct test for the situation, all values should be assessed by ANOVA then a post hoc test)

& nothing is mentioned in the footnotes!!

Authors wrote also: were compared by the student t-test or the Mann-Whitney test. : this reviewer feels different from that wrote in the table footnote , what are the basis for selecting any of the 2 tests???

Answer: The study had 2 groups. First, we do the homogeneity of variance between 2 groups. Then decided to perform student t-test or the Mann-Whitney test.

Overall the stat analysis in METHODs is not matched with that in the tables. Footnotes does not include sufficient info about the data

Answer: the footnotes have been added and revised in the tables.

Authors wrote: Pearson Chi-square test was used to compare difference between two groups.

Q: Is there a test called {Pearson Chi-square test}?????

Answer: Be honest. This is the error, we already revise it.

7. PLOS authors have the option to publish the peer review history of their article (what does this mean?). If published, this will include your full peer review and any attached files.

Do you want your identity to be public for this peer review? For information about this choice, including consent withdrawal, please see our Privacy Policy.

Reviewer #1: Yes: Gopal Nambi

Reviewer #2: No

Reviewer #3: No

---

## [Decision Letter · Decision Letter 2]

16 Apr 2021

PONE-D-20-33323R2

Effects of salbutamol on the kinetics of sevoflurane and the occurrence of early postoperative pulmonary complications in patients with mild to moderate chronic obstructive pulmonary disease: a randomized controlled study

PLOS ONE

Dear Dr. Jiang,

Thank you for submitting your manuscript to PLOS ONE. After careful consideration, we feel that it has merit but does not fully meet PLOS ONE’s publication criteria as it currently stands. Therefore, we invite you to submit a revised version of the manuscript that addresses the points raised during the review process.

We look forward to receiving your revised manuscript.

Kind regards,

Walid Kamal Abdelbasset, Ph.D.

Academic Editor

PLOS ONE

Journal Requirements:

Reviewers' comments:

Reviewer's Responses to Questions

**Comments to the Author**

1. If the authors have adequately addressed your comments raised in a previous round of review and you feel that this manuscript is now acceptable for publication, you may indicate that here to bypass the “Comments to the Author” section, enter your conflict of interest statement in the “Confidential to Editor” section, and submit your "Accept" recommendation.

Reviewer #1: All comments have been addressed

Reviewer #2: All comments have been addressed

Reviewer #3: (No Response)

2. Is the manuscript technically sound, and do the data support the conclusions?

Reviewer #1: Yes

Reviewer #2: (No Response)

Reviewer #3: Partly

3. Has the statistical analysis been performed appropriately and rigorously? 

Reviewer #1: Yes

Reviewer #2: (No Response)

Reviewer #3: No

4. Have the authors made all data underlying the findings in their manuscript fully available?

Reviewer #1: Yes

Reviewer #2: (No Response)

Reviewer #3: (No Response)

5. Is the manuscript presented in an intelligible fashion and written in standard English?

Reviewer #1: Yes

Reviewer #2: (No Response)

Reviewer #3: No

6. Review Comments to the Author

Reviewer #1: The authors have satisfactorily answered the comments and can be published in the present format. Best wishes.

Reviewer #2: The English language usage is improved. Again, reasonable people can disagree on some of the details of the statistical analysis, but these disagreements will almost certainly not change any results.

Review #3 made some suggestions for the presentation of the statistical analysis that the authors followed and that improved the manuscript.

One of the suggestions was "In methods: the difference at each time point were determined by Student’s t-tes (this is not correct to be done, t test is not the correcct test for the situation, all values should be assessed by ANOVA then a post hoc test)".

This is one of those areas where reasonable people can disagree. In fact, it is not forbidden to perform analysis separately at each time point. However, one may be concerned about the Type I error rate in that case, since it is likely that the analyses at each time point are correlated with each other. However, there is also a trade-off with the Type II error rate in that case. Note that one can also complain about the Type I error rate as it pertains to multiple testing of many different endpoints. Pragmatically, if the p-values are presented, then a reader can perform either a simple Bonferroni correction or with some effort a more elaborate correction. If the raw data are available, then clearly the reader can perform whatever analysis is felt appropriate.

Also, in fact, it is not necessary to perform an ANOVA first, followed by post hoc tests. The control of Type I error rates holds whether an ANOVA is performed first or not. This procedure (ANOVA first then post hoc contrasts second) is actually a holdover from an earlier day, when computational limitations meant that one could probably only perform Scheffe tests, in which case one wanted protection for hunting the contrast(s) responsible.

Reviewer #3: After 2 rounds of revision, authors did not transparently highlight the statistical analysis correctly. or indicate clearly the type of data in each illustration or rationalized the use of certain tests

7. PLOS authors have the option to publish the peer review history of their article (what does this mean?). If published, this will include your full peer review and any attached files.

Reviewer #1: **Yes: **Gopal Nambi

Reviewer #2: No

Reviewer #3: No

---

## [Author Response · Author response to Decision Letter 2]

21 Apr 2021

Dear editor,

Thank you for your work.

We have revised the manuscript point by point according to the reviewers’ opinion, some grammatical errors had been checked and revised. Few questions we rebut and discuss the related points in the response.

We deposited the protocols in protocols.io to enhance the reproducibility of results (dx.doi.org/10.17504/protocols.io.btidnka6).

The follows exactly showed that points (we did not change or delete the original versions you sent to us):

Thank you for your consideration. I look forward to hearing from you.

Sincerely,

Dr. Zongming Jiang

Professor and supervisor

Department of Anesthesia

Shaoxing People’s Hospital

No 568,North Zhongxing Road, Yuecheng District of Shaoxing City

Shaoxing, Zhejiang 312000, China

Tel: 86-10-575-88229212

Fax:86-10-575-88229226

E-mail: jiangzhejiang120@163.com

PONE-D-20-33323R2

Effects of salbutamol on the kinetics of sevoflurane and the occurrence of early postoperative pulmonary complications in patients with mild to moderate chronic obstructive pulmonary disease: a randomized controlled study

PLOS ONE

Dear Dr. Jiang,

Thank you for submitting your manuscript to PLOS ONE. After careful consideration, we feel that it has merit but does not fully meet PLOS ONE’s publication criteria as it currently stands. Therefore, we invite you to submit a revised version of the manuscript that addresses the points raised during the review process.

We look forward to receiving your revised manuscript.

Kind regards,

Walid Kamal Abdelbasset, Ph.D.

Academic Editor

PLOS ONE

Journal Requirements:

Answer: The reference styles has been checked according to the instruction. No retracted articles were cited as the evidence.

Reviewers' comments:

Reviewer's Responses to Questions

Comments to the Author

1. If the authors have adequately addressed your comments raised in a previous round of review and you feel that this manuscript is now acceptable for publication, you may indicate that here to bypass the “Comments to the Author” section, enter your conflict of interest statement in the “Confidential to Editor” section, and submit your "Accept" recommendation.

Reviewer #1: All comments have been addressed

Reviewer #2: All comments have been addressed

Reviewer #3: (No Response)

2. Is the manuscript technically sound, and do the data support the conclusions?

Reviewer #1: Yes

Reviewer #2: (No Response)

Reviewer #3: Partly

3. Has the statistical analysis been performed appropriately and rigorously? 

Reviewer #1: Yes

Reviewer #2: (No Response)

Reviewer #3: No

Answer: Formerly, we analyzed the main cutoff value (the FA/FI ratio), derived from the concentrations of expiratory (FA) and inspiratory (FI) fractions, and the differences at each time point were determined using Student’s t-test or the Mann-Whitney U test after one-way analysis of variance. Using the analysis method amplified the Type I error rate, a senior statistician thoroughly performed the analysis again. If all values assessed by ANOVA then conducted a post hoc test might be the reasonable way for analyzing in accordance with the data type. Intriguingly, two methods produced the similar P value and accordingly the same results. Therefore, we opt for the ANOVA then a post hoc test for analysis. In fact, we do not fully agree with the reviewer’s opinion about the statistical selection. We discussed and referred previous articles concerning the statistical analysis, ANOVA then a post hoc test for analysis might be the better method(Sturesson LW, Johansson A, Bodelsson M, Malmkvist G. Wash-in kinetics for sevoflurane using a disposable delivery system (AnaConDa) in cardiac surgery patients. Br J Anaesth. 2009 Apr;102(4):470-6. doi: 10.1093/bja/aep019. La Colla G, La Colla L, Turi S, Poli D, Albertin A, Pasculli N, Bergonzi PC, Gonfalini M, Ruggieri F. Effect of morbid obesity on kinetic of desflurane: wash-in wash-out curves and recovery times. Minerva Anestesiol. 2007 May;73(5):275-9.). St u d e n t’s t-test is also used in some papers. We provided the process of the FA/FI ratio as the supplement information.

4. Have the authors made all data underlying the findings in their manuscript fully available?

Reviewer #1: Yes

Reviewer #2: (No Response)

Reviewer #3: (No Response)

Answer: we provided the summary statistics and the data points supporting the conclusion in the submission.

5. Is the manuscript presented in an intelligible fashion and written in standard English?

Reviewer #1: Yes

Reviewer #2: (No Response)

Reviewer #3: No

Answer: Grammatical errors were already revised by WK Author Services for language proof.

6. Review Comments to the Author

Reviewer #1: The authors have satisfactorily answered the comments and can be published in the present format. Best wishes.

Reviewer #2: The English language usage is improved. Again, reasonable people can disagree on some of the details of the statistical analysis, but these disagreements will almost certainly not change any results.

Review #3 made some suggestions for the presentation of the statistical analysis that the authors followed and that improved the manuscript.

One of the suggestions was "In methods: the difference at each time point were determined by Student’s t-tes (this is not correct to be done, t test is not the correcct test for the situation, all values should be assessed by ANOVA then a post hoc test)".

This is one of those areas where reasonable people can disagree. In fact, it is not forbidden to perform analysis separately at each time point. However, one may be concerned about the Type I error rate in that case, since it is likely that the analyses at each time point are correlated with each other. However, there is also a trade-off with the Type II error rate in that case. Note that one can also complain about the Type I error rate as it pertains to multiple testing of many different endpoints. Pragmatically, if the p-values are presented, then a reader can perform either a simple Bonferroni correction or with some effort a more elaborate correction. If the raw data are available, then clearly the reader can perform whatever analysis is felt appropriate.

Also, in fact, it is not necessary to perform an ANOVA first, followed by post hoc tests. The control of Type I error rates holds whether an ANOVA is performed first or not. This procedure (ANOVA first then post hoc contrasts second) is actually a holdover from an earlier day, when computational limitations meant that one could probably only perform Scheffe tests, in which case one wanted protection for hunting the contrast(s) responsible.

Answer: Formerly, we analyzed the main cutoff value (the FA/FI ratio), derived from the concentrations of expiratory (FA) and inspiratory (FI) fractions, and the differences at each time point were determined using Student’s t-test or the Mann-Whitney U test after one-way analysis of variance. Using the analysis method amplified the Type I error rate in essence, at this time, a senior statistician thoroughly performed the analysis again. If all values assessed by ANOVA then conducted a post hoc test might be the reasonable way for analyzing in accordance with the data type. Intriguingly, two methods produced the similar P value and accordingly the same results. Therefore, we opt for the ANOVA then a post hoc test for analysis. In fact, we do not fully agree with the reviewer’s opinion about the statistical selection. We discussed and referred previous articles concerning the statistical analysis, ANOVA then a post hoc test for analysis might be the better method(Sturesson LW, Johansson A, Bodelsson M, Malmkvist G. Wash-in kinetics for sevoflurane using a disposable delivery system (AnaConDa) in cardiac surgery patients. Br J Anaesth. 2009 Apr;102(4):470-6. doi: 10.1093/bja/aep019. La Colla G, La Colla L, Turi S, Poli D, Albertin A, Pasculli N, Bergonzi PC, Gonfalini M, Ruggieri F. Effect of morbid obesity on kinetic of desflurane: wash-in wash-out curves and recovery times. Minerva Anestesiol. 2007 May;73(5):275-9.). St u d e n t’s t-test is also used in some papers. We provided the whole process of the FA/FI ratio as the supplement information.

Reviewer #3: After 2 rounds of revision, authors did not transparently highlight the statistical analysis correctly. or indicate clearly the type of data in each illustration or rationalized the use of certain tests.

Answer: we agree to your suggestion that primary outcome :FA/FI ratio , analyzed by ANOVA first then conducted a post hoc test.

7. PLOS authors have the option to publish the peer review history of their article (what does this mean?). If published, this will include your full peer review and any attached files.

Answer: The authors are pleased to publish the peer review history.

Do you want your identity to be public for this peer review? For information about this choice, including consent withdrawal, please see our Privacy Policy.

Reviewer #1: Yes: Gopal Nambi

Reviewer #2: No

Reviewer #3: No

---

## [Decision Letter · Decision Letter 3]

4 May 2021

Effects of salbutamol on the kinetics of sevoflurane and the occurrence of early postoperative pulmonary complications in patients with mild to moderate chronic obstructive pulmonary disease: a randomized controlled study

PONE-D-20-33323R3

Dear Dr. Jiang,

We’re pleased to inform you that your manuscript has been judged scientifically suitable for publication and will be formally accepted for publication once it meets all outstanding technical requirements.

Kind regards,

Walid Kamal Abdelbasset, Ph.D.

Academic Editor

PLOS ONE

Additional Editor Comments (optional):

Reviewers' comments:

Reviewer's Responses to Questions

**Comments to the Author**

1. If the authors have adequately addressed your comments raised in a previous round of review and you feel that this manuscript is now acceptable for publication, you may indicate that here to bypass the “Comments to the Author” section, enter your conflict of interest statement in the “Confidential to Editor” section, and submit your "Accept" recommendation.

Reviewer #3: All comments have been addressed

2. Is the manuscript technically sound, and do the data support the conclusions?

Reviewer #3: Yes

3. Has the statistical analysis been performed appropriately and rigorously? 

Reviewer #3: Yes

4. Have the authors made all data underlying the findings in their manuscript fully available?

Reviewer #3: No

5. Is the manuscript presented in an intelligible fashion and written in standard English?

Reviewer #3: Yes

6. Review Comments to the Author

Reviewer #3: I recommend publication of the current form of the paper titled (Effects of salbutamol on the kinetics of sevoflurane and the occurrence of early postoperative pulmonary complications in patients with mild to moderate chronic obstructive pulmonary disease: a randomized controlled study)

7. PLOS authors have the option to publish the peer review history of their article (what does this mean?). If published, this will include your full peer review and any attached files.

Reviewer #3: **Yes: **Sawsan A. Zaitone

---

## [Editor Report · Acceptance letter]

10 May 2021

PONE-D-20-33323R3 

Effects of salbutamol on the kinetics of sevoflurane and the occurrence of early postoperative pulmonary complications in patients with mild-to-moderate chronic obstructive pulmonary disease: a randomized controlled study 

Dear Dr. Jiang:

I'm pleased to inform you that your manuscript has been deemed suitable for publication in PLOS ONE. Congratulations! Your manuscript is now with our production department. 

Kind regards, 

on behalf of

Dr. Walid Kamal Abdelbasset 

Academic Editor

PLOS ONE